# Association Between the Information Environment, Knowledge, Perceived Lack of Information, and Uptake of the HPV Vaccine in Female and Male Undergraduate Students in Belgrade, Serbia

**DOI:** 10.3390/ejihpe15020021

**Published:** 2025-02-07

**Authors:** Stefan Mandić-Rajčević, Vida Jeremić Stojković, Mila Paunić, Snežana Stojanović Ristić, Marija Obradović, Dejana Vuković, Smiljana Cvjetković

**Affiliations:** 1Institute of Social Medicine, Faculty of Medicine, University of Belgrade, 11000 Belgrade, Serbia; dejana.vukovic@med.bg.ac.rs; 2School of Public Health and Health Management, Faculty of Medicine, University of Belgrade, 11000 Belgrade, Serbia; 3Department of Humanities, Faculty of Medicine, University of Belgrade, 11000 Belgrade, Serbia; vida.jeremic-stojkovic@med.bg.ac.rs (V.J.S.); smiljana.cvjetkovic@med.bg.ac.rs (S.C.); 4Institute for Students’ Health of Belgrade, 11000 Belgrade, Serbia; mila.paunic@zzzzsbg.rs (M.P.); opsta.medicina@zzzzsbg.rs (S.S.R.); ginekologija@zzzzsbg.rs (M.O.)

**Keywords:** human papilloma viruses, vaccine uptake, health behavior, infodemic, information ecosystem

## Abstract

The aim of this study was to assess the association between the use of and trust in sources of information, knowledge about human papillomavirus (HPV) and vaccines against it, perceived lack of information, and the decision to receive the HPV vaccine in undergraduate students in Belgrade. The sample of this cross-sectional study included students aged 18 to 27 who received the second dose of the HPV vaccine or used other services of the general medicine department at the Institute for Students’ Health of Belgrade during the period June–July 2024. The research instrument was a questionnaire consisting of socio-demographic data, information environment (sources of information, trust in sources of information, as well as questions related to perceived lack of information), knowledge about HPV and HPV vaccines, and vaccination status. Participants filled out an online questionnaire created on the RedCap platform of the Faculty of Medicine, University of Belgrade, which they accessed via a QR code. Hierarchical logistic regression was used to assess the association between vaccine status and socio-demographic characteristics, use and trust in information sources, knowledge, and perceived lack of information. Of the 603 participants who filled out the questionnaire completely, 78.6% were vaccinated against HPV. Key factors associated with vaccine uptake were female gender (OR = 2.33, *p* < 0.05), use of scientific literature (OR = 1.40, *p* < 0.05) and family as a source of information (OR = 1.37, *p* < 0.01), less frequent use of regional TV channels (OR = 0.76, *p* < 0.05), higher level of knowledge (OR = 1.43, *p* < 0.01), and lower perceived lack of information (OR = 0.50, *p* < 0.01). These variables explained 41% of variability in vaccine uptake in the multivariate hierarchical logistic regression model. Exposure to and trust in sources of information were significantly associated with knowledge about HPV and HPV vaccination, as well as with the perceived lack of information regarding HPV vaccination, and were the most significant determinants of the decision to accept HPV vaccine in the student population.

## 1. Introduction

With 660,000 new cases and around 350,000 deaths in 2022, cervical cancer represents the fourth most common cancer in women globally, with the highest incidence rates and mortality in low- and middle-income countries ([9]). In Europe, around 60,000 new cervical cancer cases are diagnosed annually, and in women aged 15–44, cervical cancer has the third highest incidence rate and mortality rate of 14.4 per 100,000 and 2.97 per 100,000, respectively. Serbia has the third highest age-standardized incidence rate of cervical cancer cases attributable to HPV of 18.7 per 100,000 women, surpassed by only Montenegro and Romania ([9]).

HPV infection with “high-risk” types of HPV is a major cause of cervical cancer, with persistent HPV infection being a necessary factor for invasive cervical cancer development and responsible for 73% of new cases ([11]; [10]). HPV infection is associated with almost all squamous cell carcinomas of the anus, as well as carcinomas of the vulva (15–48%), vagina (78%), penis (53%), oropharynx (13–60%), and the oral cavity, and HPV type 16 is the dominant type in these carcinomas ([14]). High-risk types, such as HPV-16 and HPV-18, are responsible for around 71% of carcinomas of the cervix, while together with HPV-45, HPV-31, HPV-33, HPV-52, and HPV-58, they are responsible for 90% of all HPV-DNA-positive squamous cell carcinomas ([15]; [51]).

Vaccination against human papillomavirus (HPV) is crucial for the primary prevention of cervical cancer, especially in low- and middle-income countries, due to the high mortality and low access to secondary and tertiary prevention ([59]). All registered vaccines are effective against HPV-16 and HPV-18, while the nonavalent vaccine also protects from HPV-31, HPV-33, HPV-45, HPV-52, and HPV-58, as well as HPV-6 and HPV-11 which cause genital warts. In the population of previously unexposed persons, the vaccine was shown to be 80–100% effective, and, in a study of over a million females, the risk of developing cervical cancer was 88% (95% CI 66–100%) lower in the group which was vaccinated before the age of 17 than in the unvaccinated group ([32]; [43]).

HPV vaccination rates vary significantly between European countries, ranging from 4% in Bulgaria to 90% in Portugal (for 2018) ([7]). Increasing HPV vaccination coverage faces many challenges globally, in Europe, and in Serbia, including lack of awareness, knowledge, and understanding of HPV infection and vaccination, concerns about vaccine safety and efficacy, perception of low risk, lack of healthcare provider recommendations, and general mistrust ([35]; [36]; [54]). Many studies have demonstrated the important association between knowledge and HPV vaccine uptake. You et al. demonstrated that enhanced awareness among female college students in China significantly improved vaccination rates ([62]). Similarly, Al-Naggar et al. reported a strong association between HPV knowledge and vaccine uptake among schoolgirls in Malaysia ([2]). Xu further highlighted that understanding the cancer prevention benefits of the HPV vaccine is critical in fostering positive vaccination attitudes and increasing uptake among adolescents and parents ([61]). A systematic review by Addisu et al. confirmed a significant association between good knowledge and higher vaccine uptake, emphasizing the importance of informed decision making ([1]). This finding was supported by Jiboc, whose systematic review revealed that better knowledge about HPV significantly enhances vaccine uptake among European women and adolescents ([24]). In Serbia, having a female child and knowledge about HPV were independently associated with a positive attitude towards vaccination among parents of children aged < 9 years ([35]). Unfortunately, Rančić et al. have shown that, 10 years from the first HPV shot being administered in Serbia, only one fifth of students knew about the existence of the HPV vaccine and only 14.2% knew about both HPV and the vaccine protecting against it ([47]). Factors such as knowledge, beliefs, and socio-economic environment have been shown to influence vaccine uptake, and the information environment is one of key factors influencing all of them and has recently been called a (social) determinant of health ([20]; [38]).

The information environment is defined as “constituted by all informational processes, services, and entities, thus including informational agents as well as their properties, interactions, and mutual relations” ([50]). “Agents” include both individuals and organizations, and they also shape and contribute to their information environment, acting alone or as a group, online or offline, as information sources ([49]). The information environment, sources of information, and trust in information play crucial roles in influencing vaccine behavior. Previous studies have found that greater trust in primary physicians, official sources of information, and governmental sources, such as the CDC, was positively associated with vaccine behavior, uptake, or intention to receive vaccines ([5]; [25]; [42]; [46]).

The HPV vaccine is primarily recommended for girls (women), aged 9 to 19 years old, and in countries with low coverage it is beneficial to also include boys (men) in HPV vaccination ([43]). In March 2022 the Health Insurance Fund of Serbia (the Fund) made the nonavalent HPV vaccine freely available to young people 9 to 19 years old, while it is not freely available to undergraduate students or anyone above the age of 19. In April 2024, the Fund made a number of doses available to undergraduate students aged 19–26 through the Institute for Students’ Health of Belgrade. During April 2024, almost 3000 students received their first shot of the nonavalent HPV vaccine and following the standard vaccine schedule (0, 2, 6 months) were to receive their second dose in June 2024. HPV vaccination provides benefits even to young adults aged 19–26, but studies on autonomous young adults who could also benefit from HPV vaccination are lacking, and factors influencing their decision could probably differ from those of parents ([12]; [34]).

The aim of this study was to evaluate the association between frequency of use of and trust in information sources, the perceived lack of information, and knowledge about HPV vaccines and the decision to get vaccinated against HPV in undergraduate students. We hypothesize that HPV vaccine uptake is associated with higher frequency of use of scientific and professional sources of HPV-related information, higher trust in that information, and higher knowledge. Also, we hypothesize that the lower perceived lack of information is associated with HPV vaccine uptake.

## 2. Material and Methods

### 2.1. Study Design and Setting

This was a cross-sectional study performed on a convenience sample of undergraduate students who use the services of the Institute for Students’ Health of Belgrade (Policlinic). The Policlinic is a dedicated healthcare institution focused on providing comprehensive primary healthcare services to students of universities and vocational schools in Belgrade.

### 2.2. Study Participants

The study sample consisted of a homogenous population of undergraduate students in Belgrade (typical start of undergraduate studies at 18–19 years), representing the target group for HPV vaccination. All participants utilized the same primary healthcare institution for students in Belgrade. The key distinction within the sample is between those who opted to get vaccinated when it became available and those who did not, despite being aware of HPV and its vaccine. In June and July 2024, those who decided to get vaccinated in April accessed the general medicine department of the Policlinic to receive their second dose of the nonavalent HPV, while other students used the services of the Policlinic for other standard reasons (e.g., current infection, regular health check-ups, referral to another healthcare facility). The recruitment was performed by study team members and student volunteers with the aim to cover the morning shift and most of the afternoon shift at the Policlinic (from 8:00 a.m. to 4:00 p.m.) every working day. All students accessing healthcare services at the department of general medicine in the study period were verbally informed about the study and provided with printed information for study participants. If they accepted to participate, they were offered a QR code which linked to the Research Electronic Data Capture (REDCap) survey hosted on secure servers of the Faculty of Medicine, University of Belgrade ([22]) .

### 2.3. Ethical Approval

The study protocol was reviewed and approved by the Ethics Committee of the Faculty of Medicine (Decision No. 25/VI-6) and the Ethics Committee of the Institute for Students’ Health of Belgrade (Decision No. 1321/2). Participants were informed about the study verbally, through printed materials, and through the electronic data collection system, and their consent to participate was a condition to continue with filling out the electronic data form.

### 2.4. Instruments

The instrument was constructed based on a literature review to include the relevant factors which compose the information environment of the study participants. It focused on information-seeking behavior ([16]; [58]), sources of information and trust in them ([57]), as well as knowledge ([45]), which could ultimately influence vaccination uptake.

The instrument included the following sections:***HPV vaccination status*** was measured by a single item assessing whether the participant took the HPV vaccine, with a binary (YES/NO) response;***Information sources’ frequency of use*** was evaluated by twelve items inquiring about the frequency of use of selected sources of information regarding HPV vaccines, on a five-point Likert scale ranging from 1 “Never” to 5 “Regularly”;***Trust in information sources*** was evaluated by twelve items inquiring about trust in selected sources of information, assessed on a five-point Likert scale ranging from 1 “Not at all trustworthy” to 5 “Completely trustworthy”;***Perceived lack of information*** was evaluated with four items on a five-point Likert scale ranging from 1 “Strongly disagree” to 5 “Strongly agree”, measuring difficulty to make a vaccination decision, confusion with information, contradictory information, and having all necessary information about the HPV vaccine. A higher score indicated a stronger feeling of a lack of information;***HPV-related knowledge*** was assessed with seven True/False questions about HPV and HPV vaccines with a “Not sure” option. The knowledge score was calculated by summing up the number of correct answers;***Socio-demographic characteristics*** included nine items: gender, age, place of birth, economic status, religiousness (scale, score from 1, indicating “Not religious at all”, to 100, indicating “Extremely religious”), study area, faculty, and marital/relationship status.

### 2.5. Pilot Testing

The instrument and the data collection system were tested in a group of students (*N* = 20) before the beginning of the study. The pilot testing was performed at the same place where the actual data collection would take place, using printed questionnaires which were filled out by the students coming to the Policlinic to receive their second HPV vaccine dose or for other reasons. The time needed to fill out the questionnaire was measured, as well as readability and understanding of all questions.

The questionnaire was estimated to take 10–15 min to fill out, which is the usual waiting time during the process of vaccination. The students found the questionnaire detailed, but understandable, with no questions representing an issue. Students also validated the list of information sources without suggestions to extend or reduce it.

During the pilot testing, a number of students (10%) had no awareness of HPV or HPV vaccines. This problem was seen in both male and female students. Considering the very specific topic and approach of the study, connected to the *decision* to get vaccinated, and the instrument, the study team decided to add a “pre-question” to the online survey which asked the participants if they are aware of the existence of HPV and HPV vaccines. In case they were completely unaware (answering “No”), the online system would block them from participating in the study and offer them the link to the webpage of the Ministry of Health and the Institute of Public Health providing information about HPV and HPV vaccination. This ensured that participants are those who were able to actually make the informed decision to get vaccinated (or not).

### 2.6. Statistical Analysis

The primary outcome (dependent variable) was HPV vaccination status (vaccinated vs. unvaccinated). The secondary outcomes were HPV-related knowledge and a perceived lack of information (continuous variables). Socio-demographic characteristics of the participants, frequency of use of information sources, trust in information sources, knowledge, and perceived lack of information were used as independent variables when the primary outcome was modeled.

Results are presented as count (%), means ± standard deviation, or median (25th–75th percentile) depending on data type and distribution of data. Groups are compared using parametric (*t*-test, ANOVA) and non-parametric (chi-square, Mann–Whitney U test, Kruskal–Wallis test) tests. Spearman’s rank correlation was used to examine the associations between the sources of information and the trust in those sources. The value of Cronbach’s alpha was calculated for the “perceived lack of information” scale which consisted of 4 items.

Linear regression was used to evaluate the association between the knowledge score and perceived lack of information (dependent variables) and socio-demographic variables, frequency of use of different sources of information about HPV vaccines, and trust thereof. The results of univariate and multivariate linear regression analysis are presented as the β coefficient and 95% confidence interval (95% CI).

Hierarchical multivariate logistic regression was performed to evaluate the association between vaccination status and independent variables. The hierarchical approach involves adding blocks of variables sequentially into the regression model to evaluate their incremental explanatory power. This allows for the assessment of how different groups of variables contribute to the prediction of the outcome variable—in this case, the likelihood of being vaccinated against HPV. Blocks were the socio-demographic characteristics of participants, frequency of use of sources of information, knowledge score, and perceived lack of information. Each group/block of variables was first used in univariate analysis, and variables identified as significantly associated with the outcome were entered into the hierarchical multivariate logistic regression. In this analysis, Nagelkerke R square was employed as a measure of the goodness-of-fit for the hierarchical multivariate regression models. Nagelkerke R square is an adjusted version of the Cox and Snell R square, designed to provide a more interpretable measure of explained variance for logistic regression models, ranging from 0 to 1 ([39]). The use of Nagelkerke R square in this context provides a clear and interpretable measure of how well each model fits the data, allowing for the comparison of the incremental contribution of each block of variables in explaining the likelihood of being vaccinated against HPV.

All *p* values less than 0.05 were considered significant. All data were analyzed using SPSS 20.0 (IBM Corp. Released 2011. IBM SPSS Statistics for Windows, Version 20.0. Armonk, NY, USA: IBM Corp.) and R 3.4.2 (R Core Team (2017). R: A language and environment for statistical computing. R Foundation for Statistical Computing, Vienna, Austria. URL https://www.R-project.org/).

## 3. Results

During the study period in June and July 2024, student volunteers estimated 1012 students used the services of the general medicine department at the Policlinic and were approached about participating in the study during their shifts. The online questionnaire was accessed by 877 students (86.6%), of which 5.2% (*N* = 46) had never heard of HPV or HPV vaccines, so they were excluded from the study and directed to the website of the Ministry of Health of Serbia and the National Institute of Public Health of Serbia website about HPV and HPV vaccines. Out of the remaining 831 students, 800 (79% of the estimated number of students who accessed the services of general medicine in the study period) accepted to participate in the study and filled out at least the socio-demographic part of the questionnaire. Six hundred and three students (75.3% of those accepting to participate in the study, 59.6% of those accessing the services in the Policlinic) completed the whole questionnaire and were included in the analysis. There were no statistically significant differences between the students who did or did not complete the whole questionnaire in gender, area of study, marital status, or income level, but students who did get vaccinated against HPV were more likely to complete the whole questionnaire (*p* < 0.001).

Table 1 shows students’ socio-demographic characteristics by vaccination status. Unvaccinated individuals were more likely to be male (31.0% vs. 14.6%) and reported higher levels of religiosity (median 68.0 vs. 55.0). In terms of field of study, a higher proportion of unvaccinated participants were from natural sciences (15.5% vs. 8.6%), whereas more vaccinated participants are from medical sciences (31.9% vs. 22.5%). Unvaccinated individuals were more often in stable relationships (47.3% vs. 35.7%), while vaccinated individuals are more frequently single (59.1% vs. 44.2%).

Table 2 presents the distribution of frequency of use of vaccine information sources among students by vaccination status. The sources of HPV-related information that were mostly relied on in our sample were internet portals (used often and regularly by 49.4%), social networks (used often and regularly by 48.3%), friends (relied on often and regularly by 45.1%), family (relied on often and regularly by 38.%), and chosen doctor (relied on often and regularly by 39.1%). Vaccinated students were more likely than unvaccinated students to often and regularly use scientific literature (30.2% vs. 15.5%, *p* < 0.001), internet portals/websites (50.4% vs. 45.8%, *p* < 0.003), family (41.3% vs. 27.2%, *p* = 0.021), and often and regularly rely on friends (48.3% vs. 33.3%, *p* = 0.005) and their doctor (42.2% vs. 27.9%, *p* = 0.009) as the source of information about vaccines.

Table 3 shows the trust in information sources by vaccination status. Scientific literature was perceived as the most credible source of information in the whole sample with 84.9% of surveyed students finding it very or completely trustworthy, followed by their chosen physician (75.8% assessed as very or completely trustworthy). More than half of responding students assessed family (56.5%) and friends (51.1%) as very or completely trustworthy sources of information. The least trusted information sources were the government (assessed as very or completely trustworthy by 6.4%) and religious leaders (assessed as very or completely trustworthy by 7.8%). Trust in scientific literature was significantly higher among vaccinated individuals, with 53.6% assessing it as completely trustworthy and 34.2% assessing it as very trustworthy, compared to 43.4% and 31.0% among the unvaccinated, respectively (*p* = 0.002). For primary physicians, 39.2% of vaccinated participants assessed their physician as completely trustworthy, and 39.0% as very trustworthy, in contrast to 35.7% and 31.0% of the unvaccinated group, respectively (*p* = 0.009). Trust in religious leaders differed significantly between vaccinated and unvaccinated participants, with only 3.8% of vaccinated participants assessing them as very trustworthy and 3.6% as completely trustworthy, while 6.2% and 3.1% of the unvaccinated express similar levels of trust (*p* = 0.019).

Table 4 shows the percentage of correct answers to questions (knowledge) about HPV and HPV vaccines by vaccination status. Vaccinated participants correctly identified that the HPV vaccine protects against oncogenic strains of the virus more frequently (90.9% vs. 57.4%, *p* < 0.001). A greater proportion of vaccinated individuals correctly understood that the vaccine is effective even if received after the first sexual encounter (80.6% vs. 46.5%, *p* < 0.001) and that it is not only for females (95.1% vs. 69.8%, *p* < 0.001). Vaccinated participants were also more aware that oncogenic strains of HPV can cause various cancers (91.4% vs. 68.2%, *p* < 0.001). While knowledge about HPV causing head and neck cancers showed no significant difference (*p* = 0.196), vaccinated individuals were more likely to correctly refute the statement that women diagnosed with HPV infection should not receive the vaccine (50.8% vs. 25.6%, *p* < 0.001) and that condom use completely prevents HPV transmission (48.3% vs. 27.1%, *p* < 0.001). Overall, the total knowledge score was higher among vaccinated participants (median score 5 vs. 3, *p* < 0.001).

Table 5 shows items related to the perceived lack of information about HPV vaccines. Among vaccinated participants, 41.1% strongly disagreed with the statement that it is difficult to decide whether to get vaccinated due to insufficient information (*p* < 0.001), 43.2% strongly disagreed with being confused by incomplete information about the HPV vaccine (*p* < 0.001), 43.2% strongly disagreed with being confused by contradictory information (*p* < 0.001), and 40.1% mostly agreed and 33.5% completely agreed they had all the necessary information about the HPV vaccine (*p* < 0.001). The perceived lack of information as a scale included four items with Cronbach’s α of 0.87. There was a statistically significant difference in the perceived lack of information between the vaccinated and unvaccinated individuals (*p* < 0.001).

Appendix A shows the univariate and multivariate logistic regressions exploring the associations between the knowledge score and perceived lack of information with the use of sources of information.

Table 6 shows the hierarchical multiple logistic regression analysis of variables significantly associated with vaccination status. The total score range for the continuous numerical variable “religiousness” was divided into quintiles for easier interpretation of the results: 0–20 (completely non-religious), 20.1–40 (moderately non-religious), 40.1–60 (neither non-religious nor religious), 60.1–80 (moderately religious), and 80.1–100 (extremely religious). Female gender (OR = 2.81, 95% CI: 1.59–4.95, *p* < 0.01) was significantly associated with higher odds of being vaccinated, while being more religious was associated with significantly lower odds of being vaccinated (OR = 0.22, 95% CI: 0.09–0.54, *p* < 0.01). The Nagelkerke R^2^ for this model was 0.145, indicating a modest proportion of variance explained by only socio-demographic factors. In Model 2, the frequencies of using scientific literature (OR = 1.70, 95% CI: 1.35–2.13, *p* < 0.01) and family as information sources (OR = 1.46, 95% CI: 1.18–1.79, *p* < 0.01) were positively associated with vaccination, while frequent use of regional TV channels was negatively associated (OR = 0.71, 95% CI: 0.56–0.90, *p* < 0.01). The significance of gender persisted, and religousness remained a significant negative predictor. Information sources about HPV vaccines further improved the model’s explanatory power (Nagelkerke R^2^ = 0.255). Model 3 incorporated knowledge about HPV and vaccines, which increased the model’s explanatory power (Nagelkerke R^2^ = 0.355). Knowledge was strongly and positively associated with vaccination status (OR = 1.68, 95% CI: 1.41–2.00, *p* < 0.01), while the effect of using scientific literature remained significant but lower (OR = 1.33, 95% CI: 1.04–1.71, *p* < 0.05). Introducing perceived lack of information in Model 4 further improved the predictive power (Nagelkerke R^2^ = 0.412). Perceived lack of information was negatively associated with being vaccinated (OR = 0.50, 95% CI: 0.38–0.67, *p* < 0.01). The positive association with knowledge remained strong, and the information sources like scientific literature and family continued to be significant predictors of vaccinations status. Gender remained a significant factor, with females still more likely to be vaccinated than males (OR = 2.33, 95% CI: 1.18–4.62, *p* < 0.05). The final model, including socio-demographic variables, information sources, knowledge, and perceived lack of information, explained 41.2% of the variance in vaccination status.

## 4. Discussion

The present study underlines the important associations between the information environment, consisting of information agents acting as information sources and the trust in them, knowledge about HPV and HPV vaccines, and perceived lack of information, in the decision to get vaccinated against HPV in a population of male and female undergraduate students. HPV vaccine uptake was most strongly associated with the female gender, lower religiousness, use of and trust in scientific literature, national TV channels, internet portals, family, friends, and chosen doctor, low exposure and trust in religious leaders, higher knowledge about HPV and HPV vaccines, and lower perceived lack of information about HPV and HPV vaccines. To our knowledge this is among the first studies exploring the influence of information environment on HPV vaccine uptake among students as representatives of young adults.

Gender was strongly associated with vaccine uptake in our study, with 85% of vaccinated students being female. In a previous study conducted in southeast Serbia, girls had a four times higher vaccination rate than boys ([47]). Similar results are seen in studies around the world where vaccination rates were 3–4 times higher in females than in males ([4]). Vaccination of females has been a priority since the first registration of the vaccine, and previous studies have found that the HPV vaccine was in much later stages of adoption among females, with over 85% who had heard of HPV vaccines, compared with males, who have shown much lower knowledge and interest, underlining much higher media coverage prioritizing females ([27]; [28]; [48]; [60]). The pilot testing of our questionnaire had shown that a number of students might be completely unaware of HPV and the HPV vaccine, and in our study that number was 45 (5.2%) of the approached students, although it is possible that some of the students who refused to participate in the study were also unaware of HPV and HPV vaccines. In a sample of college students aged 18 years and older at California State University in Los Angeles County, more than half of male and female students did not know that the HPV vaccine is recommended for their age group ([27]), which indicates a relatively high awareness (~95%) of the students in Serbia.

Religiousness, often defined as the degree of religious commitment, affiliation, beliefs, and practices within a specific religious context, can play an important role in HPV vaccine uptake. A previous study aiming at filling the identified gap in understanding the relationship between religious commitment and HPV vaccination uptake, focusing on how strong religious commitment in young adults influences vaccination decisions, found that highly religious young adults had low knowledge of HPV and HPV vaccination, and religious beliefs were associated with lower HPV vaccination uptake ([6]). Since then, studies have found religiousness to have a negative, positive, or no association with HPV vaccine uptake ([21]; [52]). In our study, religiousness, religious leaders as the source of information, and trust in religious leaders were negatively associated with knowledge about HPV, HPV vaccines, and vaccine uptake and positively associated with perceived lack of information. Nevertheless, in a recent study, Coleman et al. have indicated that single-item indices, also used in our study, although most common in the literature, may not be best suited to provide actionable insight regarding the religious beliefs/teachings which influence vaccine decisions ([13]). Therefore, a more detailed, qualitative exploration of this association is warranted.

In our study, the most used information sources about HPV and HPV vaccines were internet portals, social networks, friends and family, and a chosen (personal) doctor, while the least used information sources were religious leaders and the government. Frequent reliance on the internet and family/friends was observed in a study of Hungarian high school seniors, where healthcare professionals played only a marginal role in providing information for this study sample, which was explained by less frequent contact with the healthcare system in this age group ([3]). Results similar to ours were obtained in a survey of college students in Beijing (China), where the primary sources of HPV-related information were social media, college education, families or friends, and doctors or healthcare workers ([33]).

There was a moderate correlation between the frequency of use of specific information sources and the trust in those information sources. In fact, although surveyed students relied most frequently on information sources such as internet portals and social media, only 11.5%, and 10.2% trusted information obtained from internet portals and social media, respectively. On the other hand, scientific literature as the most trusted source is used often and regularly by less than one third of the students. UNICEF’s “Changing Childhood Project” also revealed that young people rely on social media and internet portals but do not trust them, and the greatest trust in the accuracy of information was placed on that from healthcare workers, scientists, family, and friends ([56]). This discrepancy can be explained by the fact that social media platforms are deeply ingrained in the daily lives of young people, and because they are so easily accessible, young people often rely on them out of habit or convenience, even if they have reservations about their credibility. Young people might rely on social media for quick updates but (should?) turn to more trusted sources (e.g., scientific literature, healthcare workers) for verifying important information, such as vaccine-related facts. Social media can significantly shape knowledge, attitudes, and behaviors on HPV vaccine uptake among undergraduate students. Social media platforms can enhance awareness and knowledge about HPV, as well as exposure to positive information that fosters favorable attitudes toward vaccination ([19]; [31]).

Our results show that students who were vaccinated against HPV relied on and trusted different sources of information compared to students who were not vaccinated. Using and trusting information from scientific literature, internet portals, and a chosen doctor were positively associated with higher knowledge about HPV and vaccine uptake and negatively associated with perceived lack of information to make a decision about HPV vaccination. On the other hand, using and trusting information from regional TV channels, the government, and religious leaders were negatively associated with knowledge and HPV vaccine uptake and positively associated with perceived lack of information. Vaccinated college students trusted the scientific literature, their chosen doctor, and healthcare workers from the media significantly more, while unvaccinated college students trusted religious leaders significantly more, although trust in most information sources was generally lower among unvaccinated individuals. Similar results were obtained in the study of drivers of HPV vaccine uptake among Italian university students, where students who relied on HPV-related information from healthcare workers and family members had higher inclination to get vaccinated ([12]), implying that healthcare and family settings can be very effective in driving health-related choices. Also, our findings that students who used scientific literature as a source of information more often manifested a higher level of HPV-related knowledge, felt better informed, and were more likely to get vaccinated highlight the importance of trusted, reliable sources in health education. Students who access and trust high-quality information are better equipped to make informed health decisions.

The majority of surveyed students knew that the HPV vaccine protects against oncogenic strains of human papilloma virus, that it can be received by both males and females, and that it can be effective even after the initiation of sexual activity. Regarding the diseases that can be prevented by vaccination, 86.4% new that the HPV causes cervical, vulvar, anal, penile, and oral cancer, and around one third knew that HPV can also cause some head and neck cancers. In a study of HPV-related knowledge of and attitudes towards HPV vaccination of Hungarian high school seniors, around two thirds knew that HPV can cause cervical cancer, while only small minority (below 10%) were aware of other pathologies caused by the virus in the anogenital region ([3]). Vaccinated students had higher HPV-related knowledge and felt better informed about the HPV vaccine compared to their unvaccinated counterparts. The level of HPV-related knowledge was positively associated with use of information coming from scientific literature and internet portals and negatively associated with the frequent use of national TV channels, religious leaders, and the government as information sources. Those students who more frequently relied on scientific literature felt more informed, unlike students who more frequently relied on religious leaders and YouTube channels who perceived a greater lack of information. Knowledge as a predictor of vaccination decisions was established previously ([16]; [26]) and was confirmed as a predictor of HPV vaccination in several studies ([33]; [40]; [41]), including our study.

The hierarchical multiple logistic regression analysis revealed how each additional block of variables changes the associations of previously introduced variables, while increasing the predictive power of the model. Gender and religiousness were significant predictors of vaccine uptake in the first model, but introducing information sources, such as scientific literature, regional channels, family, and religious leaders, in the second model reduced the strength of association between religiousness and vaccination uptake. In the final model, gender, scientific literature, regional channels, and family among information sources, knowledge, and perceived lack of information remained significant predictors of vaccination uptake.

Our results have demonstrated the strong relationship between the information environment, consisting of information sources and trust in them, and both knowledge and a perceived lack of information. The perceived lack of information consisted of items about difficulty in making the decision, having incomplete or contradictory information, or having all the necessary information to make the decision to get vaccinated. Those items reflect the core definition of the *infodemic*, as “An overabundance of information, some accurate and some not, that make it difficult for people to find trustworthy and reliable information in order to make a decision about their health” ([8]). Therefore, the “perceived lack of information” measured the effects of the infodemic well and was shown to be a significant predictor of the HPV vaccine uptake.

The main limitations of this study lay in its cross-sectional design and the representativeness of our sample. The cross-sectional design of this study prevents us from claiming causality in the identified strong associations between the information environment and HPV vaccine uptake, although the “exposure” to the information environment de facto happened before the decision to get vaccinated against HPV. Yet, a longitudinal design would provide a clearer picture of how knowledge and attitudes toward vaccination evolve over time. In our sample, the HPV vaccine uptake was 78.6%, which is extraordinarily high, and is much higher than the Serbian national average which is estimated to be below 4% and in reports from previous studies in Serbia where the HPV vaccine uptake was 2% ([35]). The estimated coverage in Serbia is similar to that of Turkey, with the coverage of less than 1% in girls, but much lower than that of other European countries such as Scotland (80%), Norway (73%), or Switzerland (72.6%) ([17]; [29]; [53]). The higher HPV vaccine coverage in our sample results from the fact that our study was organized during the period when students were scheduled to receive their second dose of the HPV vaccine in a short period of time, and in this period, they were the primary users of healthcare services in the Policlinic. This also explains the result that vaccinated students were significantly more likely to complete the whole questionnaire, as they might feel more responsible and even grateful that they were given the opportunity to get vaccinated against HPV for free during this period. Considering that the aim of the study was not to estimate HPV vaccine uptake but to explore the association between the information environment, knowledge, and perceived lack of information and HPV vaccine uptake, both the cross-sectional design and the present sample serve the purpose adequately. One important advantage of this sample rests on the fact that we can be fairly certain that the students’ self-reported vaccination status is correct, as they were accessing the Policlinic to receive their second dose of the HPV vaccine, which is a clear advantage compared to online surveys or in-person settings with the general student population (where we would expect to have only 2–4% vaccinated individuals). However, recruiting participants from a healthcare setting may introduce bias, as these students might already have higher awareness of HPV and vaccination, so including a broader student population might contribute to better generalizability of the results.

Future studies should dive deeper into the gender differences in the information environment, trust, knowledge, and perceived lack of information to propose gender-specific strategies to increase HPV vaccination. Qualitative methods could help understand why certain information sources are trusted, providing insights to refine communication strategies, and could also shed light on religiousness and religious beliefs and their association with HPV vaccine uptake. Further investigation into specific knowledge gaps or misconceptions about HPV and its vaccine could guide more targeted educational interventions. Additionally, examining cultural and social factors influencing trust and conducting comparative analyses across regions or countries could uncover unique trends or universal patterns to enhance global health strategies.

Our results show the predictors of HPV vaccine uptake in settings where young people can take ownership of their health and make the medical decision about vaccination, which in Serbia and many other countries include young people from the age of 15 ([37]). Methods with proven success include utilizing peer education, especially combining the peer-expert narrative ([23]), providing access to reliable information sources (websites, brochures), especially at vaccination sites or campuses ([30]), leveraging social media to share reliable information and scientific literature about HPV and vaccination ([18]), and training healthcare providers to effectively communicate the importance of the HPV vaccine ([44]), with an emphasis on health communication strategies that focus on cancer prevention benefits of the HPV vaccine which resonate well with college students ([55]).

## 5. Conclusions

The information environment and trust in information sources are significantly associated with knowledge about HPV and HPV vaccination, as well as with the perceived lack of information about HPV vaccination, a potential consequence of the infodemic. Together, they present the most significant determinants of making the decision to get vaccinated against HPV in the student population. Strategies to improve HPV vaccine uptake among adolescents and young adults should focus on using the school and college environment to promote high-quality sources of information and build trust, increase knowledge, and reduce the “perceived lack of information” caused by the infodemic.

## Figures and Tables

**Table 1 ejihpe-15-00021-t001:** Socio-demographic characteristics of study participants by vaccination status.

	All Participants	Unvaccinated	Vaccinated	*p* Value
	*N* = 603	*N* = 129	*N* = 474	
**Gender:**				<0.001
Male Female	109 (18.1%) 494 (81.9%)	40 (31.0%) 89 (69.0%)	69 (14.6%) 405 (85.4%)	
**Age (years)**	22.0 (21.0–23.0)	22.0 (20.0–23.0)	22.0 (21.0–23.0)	0.012
**Faculty group *:**				0.047
Technology and engineering sciences Sciences and mathematics Medical sciences Social sciences and humanities	125 (20.7%) 61 (10.1%) 180 (29.9%) 237 (39.3%)	29 (22.5%) 20 (15.5%) 29 (22.5%) 51 (39.5%)	96 (20.3%) 41 (8.6%) 151 (31.9%) 186 (39.2%)	
**Self-reported religiousness (1–100):**	59.0 (22.0–76.0)	68.0 (50.0–82.8)	55.0 (16.0–74.0)	<0.001
**Self-assessed financial status:**				0.128
Very good Good Average Bad Very bad I would rather not say	52 (8.6%) 225 (37.3%) 302 (50.1%) 18 (3.0%) 2 (0.3%) 4 (0.7%)	19 (14.7%) 43 (33.3%) 64 (49.6%) 3 (2.3%) 0 (0.0%) 0 (0.0%)	33 (7.0%) 182 (38.4%) 238 (50.2%) 15 (3.2%) 2 (0.4%) 4 (0.8%)	
**Relationship status:**				0.004
Single Married In a long-term relationship Other—It’s complicated I would rather not say	337 (55.9%) 3 (0.5%) 230 (38.1%) 18 (3.0%) 15 (2.5%)	57 (44.2%) 2 (1.6%) 61 (47.3%) 3 (2.3%) 6 (4.7%)	280 (59.1%) 1 (0.2%) 169 (35.7%) 15 (3.2%) 9 (1.9%)	

* Classification of faculties is based on the official organization of faculties at the University of Belgrade.

**Table 2 ejihpe-15-00021-t002:** Information sources about vaccines by vaccination status.

	All Participants	Unvaccinated	Vaccinated	*p* Value
	*N* = 603	*N* = 129	*N* = 474	
**Scientific literature:**				<0.001
Never Rarely Sometimes Often Regularly	151 (25.0%) 110 (18.2%) 179 (29.7%) 100 (16.6%) 63 (10.4%)	56 (43.4%) 20 (15.5%) 33 (25.6%) 12 (9.3%) 8 (6.2%)	95 (20.0%) 90 (19.0%) 146 (30.8%) 88 (18.6%) 55 (11.6%)	
**National TV channels:**				0.025
Never Rarely Sometimes Often Regularly	209 (34.7%) 150 (24.9%) 162 (26.9%) 61 (10.1%) 21 (3.5%)	50 (38.8%) 21 (16.3%) 33 (25.6%) 17 (13.2%) 8 (6.2%)	159 (33.5%) 129 (27.2%) 129 (27.2%) 44 (9.3%) 13 (2.7%)	
**Regional TV channels:**				0.003
Never Rarely Sometimes Often Regularly	254 (42.1%) 145 (24.0%) 142 (23.5%) 43 (7.1%) 19 (3.2%)	54 (41.9%) 21 (16.3%) 30 (23.3%) 15 (11.6%) 9 (7.0%)	200 (42.2%) 124 (26.2%) 112 (23.6%) 28 (5.9%) 10 (2.1%)	
**Internet portals:**				0.003
Never Rarely Sometimes Often Regularly	61 (10.1%) 68 (11.3%) 176 (29.2%) 192 (31.8%) 106 (17.6%)	25 (19.4%) 13 (10.1%) 32 (24.8%) 37 (28.7%) 22 (17.1%)	36 (7.6%) 55 (11.6%) 144 (30.4%) 155 (32.7%) 84 (17.7%)	
**YouTube channels:**				0.778
Never Rarely Sometimes Often Regularly	212 (35.2%) 123 (20.4%) 143 (23.7%) 93 (15.4%) 32 (5.3%)	49 (38.0%) 23 (17.8%) 32 (24.8%) 17 (13.2%) 8 (6.2%)	163 (34.4%) 100 (21.1%) 111 (23.4%) 76 (16.0%) 24 (5.1%)	
**Social networks (Instagram, Facebook, Viber, Twitter, WhatsApp):**				0.382
Never Rarely Sometimes Often Regularly	76 (12.6%) 81 (13.4%) 155 (25.7%) 170 (28.2%) 121 (20.1%)	21 (16.3%) 20 (15.5%) 33 (25.6%) 29 (22.5%) 26 (20.2%)	55 (11.6%) 61 (12.9%) 122 (25.7%) 141 (29.7%) 95 (20.0%)	
**Family:**				0.021
Never Rarely Sometimes Often Regularly	107 (17.7%) 98 (16.3%) 167 (27.7%) 147 (24.4%) 84 (13.9%)	32 (24.8%) 24 (18.6%) 38 (29.5%) 25 (19.4%) 10 (7.8%)	75 (15.8%) 74 (15.6%) 129 (27.2%) 122 (25.7%) 74 (15.6%)	
**Friends:**				0.005
Never Rarely Sometimes Often Regularly	52 (8.6%) 86 (14.3%) 193 (32.0%) 184 (30.5%) 88 (14.6%)	20 (15.5%) 20 (15.5%) 46 (35.7%) 28 (21.7%) 15 (11.6%)	32 (6.8%) 66 (13.9%) 147 (31.0%) 156 (32.9%) 73 (15.4%)	
**Your chosen physician, or physician you visit most often:**				0.009
Never Rarely Sometimes Often Regularly	113 (18.7%) 87 (14.4%) 167 (27.7%) 142 (23.5%) 94 (15.6%)	34 (26.4%) 22 (17.1%) 37 (28.7%) 17 (13.2%) 19 (14.7%)	79 (16.7%) 65 (13.7%) 130 (27.4%) 125 (26.4%) 75 (15.8%)	
**Healthcare professionals in media:**				0.606
Never Rarely Sometimes Often Regularly	125 (20.7%) 119 (19.7%) 186 (30.8%) 115 (19.1%) 58 (9.6%)	33 (25.6%) 23 (17.8%) 40 (31.0%) 22 (17.1%) 11 (8.5%)	92 (19.4%) 96 (20.3%) 146 (30.8%) 93 (19.6%) 47 (9.9%)	
**Religious leaders:**				0.020
Never Rarely Sometimes Often Regularly	498 (82.6%) 48 (8.0%) 42 (7.0%) 8 (1.3%) 7 (1.2%)	95 (73.6%) 13 (10.1%) 16 (12.4%) 3 (2.3%) 2 (1.6%)	403 (85.0%) 35 (7.4%) 26 (5.5%) 5 (1.1%) 5 (1.1%)	
**Government:**				0.265
Never Rarely Sometimes Often Regularly	379 (62.9%) 114 (18.9%) 85 (14.1%) 17 (2.8%) 8 (1.3%)	73 (56.6%) 25 (19.4%) 23 (17.8%) 6 (4.7%) 2 (1.6%)	306 (64.6%) 89 (18.8%) 62 (13.1%) 11 (2.3%) 6 (1.3%)	

**Table 3 ejihpe-15-00021-t003:** Trust in information sources about vaccines by vaccination status.

	All Participants	Unvaccinated	Vaccinated	*p* Value
	*N* = 603	*N* = 129	*N* = 474	
**Scientific literature:**				0.002
Not at all trustworthy Slightly trustworthy Moderately trustworthy Very trustworthy Completely trustworthy	5 (0.8%) 10 (1.7%) 76 (12.6%) 202 (33.5%) 310 (51.4%)	3 (2.3%) 5 (3.9%) 25 (19.4%) 40 (31.0%) 56 (43.4%)	2 (0.4%) 5 (1.1%) 51 (10.8%) 162 (34.2%) 254 (53.6%)	
**National TV channels:**				0.426
Not at all trustworthy Slightly trustworthy Moderately trustworthy Very trustworthy Completely trustworthy	166 (27.5%) 179 (29.7%) 205 (34.0%) 39 (6.5%) 14 (2.3%)	44 (34.1%) 35 (27.1%) 41 (31.8%) 6 (4.7%) 3 (2.3%)	122 (25.7%) 144 (30.4%) 164 (34.6%) 33 (7.0%) 11 (2.3%)	
**Regional TV channels:**				0.838
Not at all trustworthy Slightly trustworthy Moderately trustworthy Very trustworthy Completely trustworthy	184 (30.5%) 185 (30.7%) 190 (31.5%) 31 (5.1%) 13 (2.2%)	45 (34.9%) 38 (29.5%) 38 (29.5%) 6 (4.7%) 2 (1.6%)	139 (29.3%) 147 (31.0%) 152 (32.1%) 25 (5.3%) 11 (2.3%)	
**Internet portals:**				0.061
Not at all trustworthy Slightly trustworthy Moderately trustworthy Very trustworthy Completely trustworthy	98 (16.3%) 161 (26.7%) 275 (45.6%) 54 (9.0%) 15 (2.5%)	32 (24.8%) 32 (24.8%) 50 (38.8%) 12 (9.3%) 3 (2.3%)	66 (13.9%) 129 (27.2%) 225 (47.5%) 42 (8.9%) 12 (2.5%)	
**YouTube channels:**				0.297
Not at all trustworthy Slightly trustworthy Moderately trustworthy Very trustworthy Completely trustworthy	139 (23.1%) 176 (29.2%) 239 (39.6%) 38 (6.3%) 11 (1.8%)	37 (28.7%) 34 (26.4%) 47 (36.4%) 7 (5.4%) 4 (3.1%)	102 (21.5%) 142 (30.0%) 192 (40.5%) 31 (6.5%) 7 (1.5%)	
**Social networks (Instagram, Facebook, Viber, Twitter, WhatsApp):**				0.029
Not at all trustworthy Slightly trustworthy Moderately trustworthy Very trustworthy Completely trustworthy	110 (18.2%) 160 (26.5%) 271 (44.9%) 47 (7.8%) 15 (2.5%)	36 (27.9%) 33 (25.6%) 47 (36.4%) 10 (7.8%) 3 (2.3%)	74 (15.6%) 127 (26.8%) 224 (47.3%) 37 (7.8%) 12 (2.5%)	
**Family:**				0.738
Not at all trustworthy Slightly trustworthy Moderately trustworthy Very trustworthy Completely trustworthy	25 (4.1%) 52 (8.6%) 185 (30.7%) 199 (33.0%) 142 (23.5%)	7 (5.4%) 12 (9.3%) 43 (33.3%) 41 (31.8%) 26 (20.2%)	18 (3.8%) 40 (8.4%) 142 (30.0%) 158 (33.3%) 116 (24.5%)	
**Friends:**				0.276
Not at all trustworthy Slightly trustworthy Moderately trustworthy Very trustworthy Completely trustworthy	18 (3.0%) 37 (6.1%) 246 (40.8%) 218 (36.2%) 84 (13.9%)	7 (5.4%) 8 (6.2%) 55 (42.6%) 46 (35.7%) 13 (10.1%)	11 (2.3%) 29 (6.1%) 191 (40.3%) 172 (36.3%) 71 (15.0%)	
**Your chosen physician, or physician you visit most often:**				0.009
Not at all trustworthy Slightly trustworthy Moderately trustworthy Very trustworthy Completely trustworthy	8 (1.3%) 19 (3.2%) 119 (19.7%) 225 (37.3%) 232 (38.5%)	5 (3.9%) 7 (5.4%) 31 (24.0%) 40 (31.0%) 46 (35.7%)	3 (0.6%) 12 (2.5%) 88 (18.6%) 185 (39.0%) 186 (39.2%)	
**Healthcare professionals in media:**				0.003
Not at all trustworthy Slightly trustworthy Moderately trustworthy Very trustworthy Completely trustworthy	56 (9.3%) 108 (17.9%) 221 (36.7%) 131 (21.7%) 87 (14.4%)	23 (17.8%) 19 (14.7%) 48 (37.2%) 21 (16.3%) 18 (14.0%)	33 (7.0%) 89 (18.8%) 173 (36.5%) 110 (23.2%) 69 (14.6%)	
**Religious leaders:**				0.019
Not at all trustworthy Slightly trustworthy Moderately trustworthy Very trustworthy Completely trustworthy	401 (66.5%) 75 (12.4%) 80 (13.3%) 26 (4.3%) 21 (3.5%)	72 (55.8%) 18 (14.0%) 27 (20.9%) 8 (6.2%) 4 (3.1%)	329 (69.4%) 57 (12.0%) 53 (11.2%) 18 (3.8%) 17 (3.6%)	
**Government:**				0.633
Not at all trustworthy Slightly trustworthy Moderately trustworthy Very trustworthy Completely trustworthy	340 (56.4%) 109 (18.1%) 115 (19.1%) 22 (3.6%) 17 (2.8%)	71 (55.0%) 21 (16.3%) 30 (23.3%) 3 (2.3%) 4 (3.1%)	269 (56.8%) 88 (18.6%) 85 (17.9%) 19 (4.0%) 13 (2.7%)	

Frequency of use and trust in the respective information sources were moderately correlated (Spearman ρ between 0.5 and 0.6). After testing with both sources of information and trust in the sources of information, trust was excluded from the hierarchical multiple logistic regression analysis due to collinearity and providing no additional value to the model.

**Table 4 ejihpe-15-00021-t004:** Knowledge about HPV and HPV vaccine by vaccination status.

	All Participants	Unvaccinated	Vaccinated	*p* Value
	*N* = 603	*N* = 129	*N* = 474	
**The HPV vaccine protects against oncogenic (cancer-causing) strains of the human papillomavirus.**				<0.001
Correct * Incorrect Not sure	505 (83.7%) 12 (2.0%) 86 (14.3%)	74 (57.4%) 7 (5.4%) 48 (37.2%)	431 (90.9%) 5 (1.1%) 38 (8.0%)	
**The HPV vaccine is only effective if it is received before the first sexual intercourse.**				<0.001
Correct Incorrect * Not sure	33 (5.5%) 442 (73.3%) 128 (21.2%)	15 (11.6%) 60 (46.5%) 54 (41.9%)	18 (3.8%) 382 (80.6%) 74 (15.6%)	
**The HPV vaccine is intended for females only.**				<0.001
Correct Incorrect * Not sure	16 (2.7%) 541 (89.7%) 46 (7.6%)	10 (7.8%) 90 (69.8%) 29 (22.5%)	6 (1.3%) 451 (95.1%) 17 (3.6%)	
**Oncogenic strains of human papillomavirus can cause cancer of the cervix, vagina, vulva, penis, anus, and oral cavity.**				<0.001
Correct * Incorrect Not sure	521 (86.4%) 11 (1.8%) 71 (11.8%)	88 (68.2%) 7 (5.4%) 34 (26.4%)	433 (91.4%) 4 (0.8%) 37 (7.8%)	
**HPV causes some cancers of the head and neck.**				0.196
Correct * Incorrect Not sure	174 (28.9%) 96 (15.9%) 333 (55.2%)	29 (22.5%) 22 (17.1%) 78 (60.5%)	145 (30.6%) 74 (15.6%) 255 (53.8%)	
**Females diagnosed with HPV infection should not receive the HPV vaccine.**				<0.001
Correct Incorrect * Not sure	63 (10.4%) 274 (45.4%) 266 (44.1%)	17 (13.2%) 33 (25.6%) 79 (61.2%)	46 (9.7%) 241 (50.8%) 187 (39.5%)	
**The use of condoms prevents the transmission of HPV.**				<0.001
Correct Incorrect * Not sure	175 (29.0%) 264 (43.8%) 164 (27.2%)	39 (30.2%) 35 (27.1%) 55 (42.6%)	136 (28.7%) 229 (48.3%) 109 (23.0%)	
**Total score (no. correct)**	5.0 (0.0–7.0)	3.0 (0.0–7.0)	5.0 (0.0–7.0)	<0.001

* indicates the correct answer.

**Table 5 ejihpe-15-00021-t005:** Perceived lack of information by vaccination status.

	All Participants	Unvaccinated	Vaccinated	*p* Value
	*N* = 603	*N* = 129	*N* = 474	
**It is hard to make the decision whether to vaccinate against HPV, since there is a lack of information about vaccines.**				<0.001
Strongly disagree Disagree Neither disagree nor agree Agree Strongly agree	220 (36.5%) 138 (22.9%) 121 (20.1%) 87 (14.4%) 37 (6.1%)	25 (19.4%) 17 (13.2%) 38 (29.5%) 27 (20.9%) 22 (17.1%)	195 (41.1%) 121 (25.5%) 83 (17.5%) 60 (12.7%) 15 (3.2%)	
**Incomplete information regarding the HPV vaccines I come across make me confused.**				<0.001
Strongly disagree Disagree Neither disagree nor agree Agree Strongly agree	225 (37.3%) 146 (24.2%) 135 (22.4%) 64 (10.6%) 33 (5.5%)	20 (15.5%) 19 (14.7%) 48 (37.2%) 26 (20.2%) 16 (12.4%)	205 (43.2%) 127 (26.8%) 87 (18.4%) 38 (8.0%) 17 (3.6%)	
**Contradictory information regarding the HPV vaccines I come across make me confused.**				<0.001
Strongly disagree Disagree Neither disagree nor agree Agree Strongly agree	229 (38.0%) 142 (23.5%) 138 (22.9%) 58 (9.6%) 36 (6.0%)	24 (18.6%) 21 (16.3%) 47 (36.4%) 21 (16.3%) 16 (12.4%)	205 (43.2%) 121 (25.5%) 91 (19.2%) 37 (7.8%) 20 (4.2%)	
**I have all the information I need regarding HPV vaccination.**				<0.001
Strongly disagree Disagree Neither disagree nor agree Agree Strongly agree	34 (5.6%) 52 (8.6%) 121 (20.1%) 216 (35.8%) 180 (29.9%)	20 (15.5%) 21 (16.3%) 41 (31.8%) 26 (20.2%) 21 (16.3%)	14 (3.0%) 31 (6.5%) 80 (16.9%) 190 (40.1%) 159 (33.5%)	
**Perceived lack of information (scale)**	2.0 (1.2–3.0)	3.0 (2.2–3.5)	2.0 (1.2–2.8)	<0.001

**Table 6 ejihpe-15-00021-t006:** Hierarchical multiple logistic regression analysis of variables significantly associated with vaccination status.

Variable	Model 1 ^a^	Model 2 ^b^	Model 3 ^c^	Model 4 ^d^
	OR (95% CI)	OR (95% CI)	OR (95% CI)	OR (95% CI)
**Gender**				
Male Female	Ref. 2.81 (1.59–4.95) **	Ref. 2.66 (1.42–4.99) **	Ref. 2.02 (1.04–3.91) *	Ref. 2.33 (1.18–4.62) *
**Age**	1.17 (1.03–1.32) *	1.17 (1.02–1.34) *	1.03 (0.89–1.19)	1.05 (0.90–1.22)
**Religiousness**				
Completely non-religious Moderately non-religious Neither non-religious nor religious Moderately religious Extremely religious	Ref. 0.77 (0.26–2.31) 0.37 (0.15–0.94) * 0.32 (0.13–0.77) * 0.22 (0.09–0.54) **	Ref. 0.78 (0.25–2.46) 0.40 (0.15–1.06) 0.37 (0.15–0.93) * 0.28 (0.11–0.75) *	Ref. 0.81 (0.25–2,64) 0.48 (0.18–1.31) 0.46 (0.18–1.20) 0.31 (0.12–0.83) *	Ref. 0.89 (0.26–3.02) 0.52 (0.19–1.43) 0.55 (0.21–1.47) 0.38 (0.14–1.06)
**Faculty group *:**				
Technology and engineering sciences Sciences and mathematics Medical sciences Social sciences and humanities	Ref. 0.41 (0.17–0.95) * 1.41 (0.71–2.77) 0.81 (0.43–1.51)	Ref. 0.40 (0.16–0.97) * 0.90 (0.42–1.93) 0.77 (0.40–1.47)	Ref. 0.37 (0.14–0.93) * 0.84 (0.38–1.88) 0.91 (0.45–1.81)	0.34 (0.13–0.91) * 0.60 (0.26–1.40) 0.92 (0.45–1.88)
**Relationship status**				
Single Married In a long-term relationship Other	Ref. 0.11 (0.01–1.45) 0.56 (0.35–0.89) * 1.24 (0.23–6.76)	Ref. 0.11 (0.01–1.62) 0.59 (0.36–0.96) * 1.45 (0.25–8.25)	Ref. 0.18 (0.01–2.27) 0.60 (0.36–1.01) 3.20 (0.48–21.56)	Ref. 0.22 (0.02–2.87) 0.48 (0.28–0.85) * 3.34 (0.49–22.61)
**Scientific literature (frequency of use)**		1.70 (1.35–2.13) **	1.33 (1.04–1.71) *	1.40 (1.08–1.82) *
**Regional channels ( frequency of use)**		0.71 (0.56–0.90) **	0.79 (0.62–1.01)	0.76 (0.59–0.99) *
**Family** **(frequency of use)**		1.46 (1.18–1.79) **	1.43 (1.15–1.78) *	1.37 (1.08–1.72) **
**Religious leaders** **(frequency of use)**		0.67 (0.48–0.94) *	0.90 (0.62–1.31)	0.89 (0.61–1.31)
**Knowledge (score)**			1.68 (1.41–2.00) **	1.43 (1.19–1.72) **
**Perceived lack of information (score)**				0.50 (0.38–0.67) **
**Nagelkerke R^2^**	0.145	0.255	0.355	0.412

* *p* < 0.05; ** *p* < 0.01; ^a^ Model 1 includes socio-demographic variables; ^b^ Model 2 adds information sources about HPV vaccines; ^c^ Model 3 adds knowledge about HPV and vaccines; ^d^ Model 4 adds perceived lack of information.

## Data Availability

Data available upon reasonable request to the corresponding author.

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
