# Peer review of "Association Between the Information Environment, Knowledge, Perceived Lack of Information, and Uptake of the HPV Vaccine in Female and Male Undergraduate Students in Belgrade, Serbia"

_ejihpe, 2025, doi:10.3390/ejihpe15020021_

Round 1

Reviewer 1 Report

Comments and Suggestions for Authors

This study explores the relationship between the information environment, sources of information, and behavior, presenting several interesting findings. However, several issues need to be addressed before the manuscript can be considered for acceptance.

1. Hypotheses and Conceptual Framework
While the authors provide a theoretical foundation regarding the information environment, sources of information, trust, and vaccine behavior, they do not seem to propose hypotheses regarding their interactions before testing them. Stating the hypotheses explicitly and providing a conceptual framework (e.g., a diagram) to clarify how these elements interact would significantly improve the study's coherence and methodological rigor.

2. Sample Representativeness
The sample appears unrepresentative based on the current description: it includes only participants coming for the second dose, those already aware of HPV and the HPV vaccine, and those with some knowledge. This could lead to selection bias and limit the generalizability of the findings.

3. Results Presentation
The results section is overly detailed and does not sufficiently focus on the associations between the information environment, sources of information, trust, and vaccine uptake. Reducing the emphasis on individual domains and improving the reporting of these key associations would enhance the study’s impact and relevance.

4. Discussion Alignment
The discussion section does not consistently rely on the study’s findings. Stronger alignment with the results is necessary to support the statements made.

Major Issues:

  1. Introduction:

- The cost of the HPV vaccine might be a critical factor, especially if only a limited number of free doses are available to undergraduate students. And who gets free vaccine if limited number is available?

- The authors stated that the aim is to “evaluate the association between exposure to and trust in information sources, the perceived lack of information, and knowledge about HPV vaccines and the decision to get vaccinated against HPV in undergraduate students”. However, they excluded those without exposure in the first place. The sample seems to be biased towards those with exposure already.

  1. Materials and Methods:

Study Participants: The inclusion of only those coming for the second dose may bias the sample towards pro-vaccine individuals. Justification for this criterion is needed.

Pilot Testing: Excluding individuals who were completely unaware of HPV or the HPV vaccine is not sufficiently justified and may skew the findings.

Statistical Analysis: Structural Equation Modeling (SEM) may be more appropriate than hierarchical multivariate logistic regression for testing complex relationships. Additionally, the authors should check for multicollinearity between blocks before including them in the model.

  1. Results:

- Those never heard of HPV or HPV vaccine (5.2%) were excluded. Were their characteristics statistically significantly different from those included?

- Among those 603 participants who completed the questionnaire, no missing value at all? If not, how the authors deal with missing value?

- The claim that vaccinated students were more likely to complete the entire questionnaire suggests the sample may not be representative. This limitation should be addressed.

- If knowledge scores, sources of information, and perceived lack of information are associated, did the authors test for interactions among these variables? How do these factors collectively or independently influence vaccine uptake?

  1. Discussion:

Lines 60–61, 83–84, and 134–135: The authors make several claims that lack clear support from the study’s findings. Please provide specific results to substantiate these statements.

Author Response

### We thank the reviewer for the time and effort put into the revision of our Manuscript. We have done our best to answer all the issues, provide a rationale for our decisions, and indicate where changes have been made in the text.

This study explores the relationship between the information environment, sources of information, and behavior, presenting several interesting findings. However, several issues need to be addressed before the manuscript can be considered for acceptance.

  1. Hypotheses and Conceptual Framework
    While the authors provide a theoretical foundation regarding the information environment, sources of information, trust, and vaccine behavior, they do not seem to propose hypotheses regarding their interactions before testing them. Stating the hypotheses explicitly and providing a conceptual framework (e.g., a diagram) to clarify how these elements interact would significantly improve the study's coherence and methodological rigor.

### Thank you for this comment and proposal of using hypotheses. Although we agree that hypothesis setting and testing is useful in various scenarios, we believe there are situations where they might not be crucial or extremely useful, and this is often seen in studies of drivers of health behavior, due to the complexity of health behaviors, the influence of the context, and forcing a narrow focus on an explorative study. We do, however agree that using hypotheses might help the reader better understand the scope of the research, so we have added three hypotheses.

--- Changes in text: Introduction, final paragraph. The aim of this study was to evaluate the association between  frequency of use of and trust in information sources, the perceived lack of information, and knowledge about HPV vaccines and the decision to get vaccinated against HPV in undergraduate stu-dents. We hypothesize that HPV vaccine acceptance is associated with higher frequency of use of scientific and professional sources of HPV-related information, higher trust in that information, and higher knowledge. Also, we also hypothesize that the lower perceived lack of information is associated with HPV vaccine acceptance.

  1. Sample Representativeness
    The sample appears unrepresentative based on the current description: it includes only participants coming for the second dose, those already aware of HPV and the HPV vaccine, and those with some knowledge. This could lead to selection bias and limit the generalizability of the findings.

### The sample includes a homogenous population of undergraduate students studying in Belgrade. This is very representative of that specific target population for HPV vaccination: they are all students studying in the same city, using the services of the same main/primary healthcare institution for students in Belgrade. Their main difference is that some have decided to get vaccinated once the opportunity was available, and others did not (but were aware of HPV and HPV vaccination). To better explain the context, HPV vaccination is free and available since 2022 to all boys and girls from 9 years old to 19 years old (18-19 is when undergraduate studies start in Serbia). Knowledge was not a prerequisite to participate in the study, only knowing of HPV and HPV-vaccine – since our focus was on students who we believe were able to make a decision to get vaccinated (before they turned 20, or by using the opportunity which was provided to them in April 2024 by the government). We have tried to clarify this context better in the Methods section and in the limitations of the study.

--- Changes in text: in Introduction, Paragraph 6; in Methods, section 2.2. Study participants; in Discussion, Paragraph 10 – Limitations.

  1. Results Presentation
    The results section is overly detailed and does not sufficiently focus on the associations between the information environment, sources of information, trust, and vaccine uptake. Reducing the emphasis on individual domains and improving the reporting of these key associations would enhance the study’s impact and relevance.

### Thank you for this comment, and we have discussed within our team and we agree that too much space is given to individual domains, so we have moved tables 6 and 7 to Supplementary material and we provide only a small link with this additional information in the original text, while focusing more on the primary aims of the study.

--- Changes in text: Tables 6 and 7 with corresponding description moved to Supplementary Material 1.

  1. Discussion Alignment
    The discussion section does not consistently rely on the study’s findings. Stronger alignment with the results is necessary to support the statements made.

 ### Thank you for this comment. We have focused the discussion more on the study’s findings and aligned it by reducing parts which were only slightly connected to our results. We believe the Discussion is shorter and easier to read now.

--- Changes in text: Discussion, Paragraphs 4, 5, 8, 10.

Major Issues:

  1. Introduction:

- The cost of the HPV vaccine might be a critical factor, especially if only a limited number of free doses are available to undergraduate students. And who gets free vaccine if limited number is available?

### We explained above the pricing and cost, but we would like to underline that all students who applied actually got the vaccine. The number was limited in the sense that it was projected to cover assessed needs of student population. We excluded the word “limited” to avoid misunderstandings. Also, our improvement of the Methods and Introduction should better resolve this issue.

--- Changes in text: in Introduction, Paragraph 6; in Methods, section 2.2. Study participants.

- The authors stated that the aim is to “evaluate the association between exposure to and trust in information sources, the perceived lack of information, and knowledge about HPV vaccines and the decision to get vaccinated against HPV in undergraduate students”. However, they excluded those without exposure in the first place. The sample seems to be biased towards those with exposure already.

### Thank you for this comment and this opened an interesting discussion in our team. Our focus was the decision to get vaccinated, and in our study, we focused on those who were able to make that decision (meaning they were at least aware of HPV and vaccine against it).

--- Changes in text: In Introduction - We reworded this sentence to be more precise: “The aim of this study was to evaluate the association between frequency of use of and trust in information sources, the perceived lack of information, and knowledge about HPV vaccines and the decision to get vaccinated against HPV in undergraduate students.“

  1. Materials and Methods:

Study Participants: The inclusion of only those coming for the second dose may bias the sample towards pro-vaccine individuals. Justification for this criterion is needed.

### Thank you for this comment. We tried to clarify that one part of the sample did NOT include only those coming for the second dose, but both those who were offered the chance to get vaccinated and decided to do so (in April 2024, coming for the second dose in June/July 2024), and their counterparts who knew about HPV and HPV vaccine, but did not do so (coming to the Policlinic for all the other reasons). It is clear that those who were vaccinated might be considered PRO vaccine, while the others are on the “spectrum”, and our study explores their differences in information environment, knowledge and perceived lack of information.

--- Changes in text: in Introduction and Methods (study participants section).

Pilot Testing: Excluding individuals who were completely unaware of HPV or the HPV vaccine is not sufficiently justified and may skew the findings.

### We were exploring the decision to get vaccinated and reasoned among our team that a person unaware of HPV and HPV vaccine cannot be considered as a person who “made a decision” NOT to get vaccinated. We were specifically aiming at those who could (the vaccine was available, they were aware of HPV and vaccine against it) but decided not to get vaccinated. Unfortunately, at the time of the study design, we did NOT opt to collect information about those who were excluded due to not being aware of HPV and vaccine against it.

Statistical Analysis: Structural Equation Modeling (SEM) may be more appropriate than hierarchical multivariate logistic regression for testing complex relationships. Additionally, the authors should check for multicollinearity between blocks before including them in the model.

### Thank you for this comment, and we have consulted our resident statistician regarding these issues before opting for our approach. We have decided to apply hierarchical logistic regression since it is well-suited for identifying predictors of a binary outcome, such as vaccination behavior (vaccinated vs. not vaccinated). Its primary strength lies in assessing the direct relationship between independent variables and the dependent variable, making it ideal for studies aiming to determine which factors significantly predict vaccination behavior. We did check for multicollinearity between obvious suspects, such as use of information sources and trust in information sources, therefore trust was excluded from the hierarchical logistic regression analysis.

  1. Results:

- Those never heard of HPV or HPV vaccine (5.2%) were excluded. Were their characteristics statistically significantly different from those included?

### This is a very good question we have asked ourselves (too late), but unfortunately this was not in our focus when we designed the study – we have opted NOT to collect additional information about those who were excluded and therefore we cannot comment on this.

- Among those 603 participants who completed the questionnaire, no missing value at all? If not, how the authors deal with missing value?

### The RedCap system was programmed to avoid missing values – participants were not allowed to move on to the next portion of questions if the previous one was not completed. Our volunteers were present to help in case there were problematic questions.

- The claim that vaccinated students were more likely to complete the entire questionnaire suggests the sample may not be representative. This limitation should be addressed.

### Thank you for this comment and we are aware of this limitation. We have openly discussed the representativeness of our sample considering this limitation in the “Limitations” paragraph of the Discussion (Paragraph 10).

--- In text: The higher HPV vaccine coverage in our sample results from the fact that our study was organized during the period when students were scheduled to receive their second dose of the HPV vaccine in a short period of time, and in this period, they were the primary users of healthcare services in the Policlinic. This also explains the result that vaccinated students were significantly more likely to complete the whole questionnaire, as they might feel more responsible and even grateful that they were given the opportunity to get vaccinated against HPV for free during this period.

- If knowledge scores, sources of information, and perceived lack of information are associated, did the authors test for interactions among these variables? How do these factors collectively or independently influence vaccine uptake?

### After consultations with our resident statistician, we opted for individually testing for associations and including those variables proven significantly associated in the multivariate hierarchical regression model. Our reasoning was to use the most straight forward method being an explorative study, acknowledging the lack of a good/existing framework connecting the information environment and health behavior, and our sample size.

  1. Discussion:

Lines 60–61, 83–84, and 134–135: The authors make several claims that lack clear support from the study’s findings. Please provide specific results to substantiate these statements.

### Thank you for pointing out these claims. We deleted these sentences.

Reviewer 2 Report

Comments and Suggestions for Authors

Introduction

This study explores the link between information sources, trust in those sources, knowledge about HPV and its vaccines, and the decision to get vaccinated among undergraduate students in Belgrade. By focusing on a crucial age group, the research addresses an important aspect of public health: understanding the factors that influence HPV vaccination rates.

Strengths

The study tackles a significant public health issue—HPV vaccination uptake—among young adults, a key demographic for HPV prevention. Its findings contribute to the global discussion on improving vaccine acceptance, especially in regions with varying levels of coverage.

The use of a comprehensive questionnaire to gather socio-demographic data, sources of information, and knowledge levels ensures that the research covers multiple dimensions. The hierarchical logistic regression approach adds depth to the analysis by accounting for various influencing factors.

With 603 completed responses, the sample size is substantial, providing reliable data and reducing the risk of statistical error. This enhances the study's credibility and allows for meaningful conclusions about the target population.

The statistical methods were very well employed, with adequate test-testing for comparisons between groups. The adjustments were made using multivariate analysis, which appears to have been very well conducted. In this sense, I have just one question: was correction for multiple comparisons made? For example, Bonferroni correction.

The identification of factors such as gender, trust in family, and the use of scientific literature as significant predictors of vaccine acceptance is particularly valuable. The finding that higher knowledge correlates with increased vaccination rates underscores the importance of education and awareness campaigns.

Utilizing an online questionnaire accessed via QR codes demonstrates a modern and efficient approach to data collection. This method likely increased accessibility and participation among tech-savvy students.

Areas for Improvement

Since the study is cross-sectional, it can only show associations, not causal relationships. A longitudinal design would provide a clearer picture of how knowledge and attitudes toward vaccination evolve over time.

Recruiting participants from a healthcare setting may introduce bias, as these students might already have higher health awareness. Including a broader student population from different university departments could improve the generalizability of the results.

While the study highlights which information sources are trusted, a qualitative component could provide insights into why certain sources are more influential. Understanding the reasons behind this trust would help tailor more effective communication strategies.

Breaking down the knowledge component to identify specific gaps or misconceptions about HPV and its vaccine could enhance educational interventions. This would allow for more targeted messaging.

The study could benefit from a deeper exploration of cultural or social factors that influence trust in information sources. Comparative analyses with other regions or countries could help identify unique trends or universal patterns.

Conclusion

Overall, this study offers valuable insights into the factors influencing HPV vaccine acceptance among undergraduate students. Its findings have important implications for public health initiatives aimed at increasing vaccination rates. By addressing the limitations and expanding the scope of future research, these insights could be further enhanced to inform more effective strategies for promoting HPV vaccination.

Author Response

### We would like to thank the Reviewer for the time and effort put into the revision of our Manuscript. It is also positive to read the identified strength of our work, although we are well aware that our work has several limitations. We have done our best to answer all the issues the Reviewer pointed out, provide a rationale for our decisions, and indicate where changes have been made in the text.

Introduction

This study explores the link between information sources, trust in those sources, knowledge about HPV and its vaccines, and the decision to get vaccinated among undergraduate students in Belgrade. By focusing on a crucial age group, the research addresses an important aspect of public health: understanding the factors that influence HPV vaccination rates.

Strengths

The study tackles a significant public health issue—HPV vaccination uptake—among young adults, a key demographic for HPV prevention. Its findings contribute to the global discussion on improving vaccine acceptance, especially in regions with varying levels of coverage.

The use of a comprehensive questionnaire to gather socio-demographic data, sources of information, and knowledge levels ensures that the research covers multiple dimensions. The hierarchical logistic regression approach adds depth to the analysis by accounting for various influencing factors.

With 603 completed responses, the sample size is substantial, providing reliable data and reducing the risk of statistical error. This enhances the study's credibility and allows for meaningful conclusions about the target population.

The statistical methods were very well employed, with adequate test-testing for comparisons between groups. The adjustments were made using multivariate analysis, which appears to have been very well conducted. In this sense, I have just one question: was correction for multiple comparisons made? For example, Bonferroni correction.

### Thank you for your comment and proposal. We discussed the use of a correction with our resident statistician and opted not to use it for the following reasoning. The Bonferroni adjustment is often avoided in health behavior and exploratory studies due to its overly conservative nature, which increases the risk of Type II errors, potentially overlooking true effects, especially in small samples or modest effect sizes. Additionally, it assumes independence among tests, which is rarely the case in studies of health outcomes, leading to an underestimation of significance. This is why we have, after consideration, opted not to use any kind of correction.

The identification of factors such as gender, trust in family, and the use of scientific literature as significant predictors of vaccine acceptance is particularly valuable. The finding that higher knowledge correlates with increased vaccination rates underscores the importance of education and awareness campaigns.

Utilizing an online questionnaire accessed via QR codes demonstrates a modern and efficient approach to data collection. This method likely increased accessibility and participation among tech-savvy students.

### Thank you for this comment. Our students volunteers were present during the data collection and we have tested the use of the QR code, which was proven not to be a problem among our study population.

Areas for Improvement

Since the study is cross-sectional, it can only show associations, not causal relationships. A longitudinal design would provide a clearer picture of how knowledge and attitudes toward vaccination evolve over time.

### We included this suggestion in the Limitations section (lines 153-155).

Recruiting participants from a healthcare setting may introduce bias, as these students might already have higher health awareness. Including a broader student population from different university departments could improve the generalizability of the results.

### We added this in the limitations section (lines 175-178).

While the study highlights which information sources are trusted, a qualitative component could provide insights into why certain sources are more influential. Understanding the reasons behind this trust would help tailor more effective communication strategies.

Breaking down the knowledge component to identify specific gaps or misconceptions about HPV and its vaccine could enhance educational interventions. This would allow for more targeted messaging.

The study could benefit from a deeper exploration of cultural or social factors that influence trust in information sources. Comparative analyses with other regions or countries could help identify unique trends or universal patterns.

 ### Thank you for these suggestions. We have rewritten our Limitations and Future studies paragraph to reflect these suggestions which we fully support.

--- Changes in text: Future studies should dive deeper into the gender differences in the information environment, trust, knowledge and perceived lack of information to propose gender-specific strategies to increase HPV vaccination. Qualitative methods could help understand why certain information sources are trusted, providing insights to refine communication strategies, and could also shed light on religiousness and religious beliefs and their association with HPV vaccine uptake. Further investigation into specific knowledge gaps or mis-conceptions about HPV and its vaccine could guide more targeted educational interventions. Additionally, examining cultural and social factors influencing trust and conducting comparative analyses across regions or countries could uncover unique trends or universal patterns to enhance global health strategies.

Conclusion

Overall, this study offers valuable insights into the factors influencing HPV vaccine acceptance among undergraduate students. Its findings have important implications for public health initiatives aimed at increasing vaccination rates. By addressing the limitations and expanding the scope of future research, these insights could be further enhanced to inform more effective strategies for promoting HPV vaccination.

### We would like to thank the reviewer again for their kind words and constructive suggestions which we have adopted fully.

Reviewer 3 Report

Comments and Suggestions for Authors

There may be fundamental errors in the research design, questions, or methods. For example, how could the age of undergraduates is range from 18 to 27 (methods section).

There are too many of inconsistencies throughout the draft, such as in abstract the age range is 19-26 but in the main body 18-27.

The study relies on self-reported vaccination status, which may be subject to social desirability bias or recall bias.

While the study identifies religiosity as a factor influencing vaccine uptake, it does not delve into the specific religious teachings or beliefs that may influence this decision. A more nuanced understanding of the intersection between religious beliefs and health behavior is needed.

The language writings make the manuscript hard to follow. It should be refined before any further evaluations.

Comments on the Quality of English Language

The English in the draft does indeed require substantial revision to meet the standards expected in academic publishing.

Author Response

### We thank the reviewer for the time and effort put into the revision of our Manuscript. We have done our best to answer all the issues, provide a rationale for our decisions, and indicate where changes have been made in the text.

There may be fundamental errors in the research design, questions, or methods. For example, how could the age of undergraduates is range from 18 to 27 (methods section).

### Undergraduate studies in Serbia start at the age of 18-19 and are the first level studies after high school. The law on health protection of Serbia defines this population as vulnerable granting additional provisions (such as free healthcare) until the end of the studies or by the age of 26. We have expanded the age to include both 18 and 27 to include those who have just turned 27 during the year of the study and those who might have started school earlier and were 18 when they started undergraduate studies.

--- Changes in text: In Methods, Study Participants: The study sample consisted of a homogenous population of undergraduate students in Belgrade (typical start of undergraduate studies at 18–19 years), representing the target group for HPV vaccination. All participants utilized the same primary healthcare institu-tion for students in Belgrade. The key distinction within the sample is between those who opted to get vaccinated when it became available and those who did not, despite being aware of HPV and its vaccine.

There are too many of inconsistencies throughout the draft, such as in abstract the age range is 19-26 but in the main body 18-27.

### Thank you for pointing this out, we have corrected the error in the Abstract.

--- Changes in text: in Abstract.

The study relies on self-reported vaccination status, which may be subject to social desirability bias or recall bias.

### Thank you for this point. We have actively worked to reduce this bias, and this is actually one of the strengths of our study. We avoided this bias by including students who were accessing the Policlinic to receive their second dose of the HPV vaccine, so can be fairly certain that the students’ self-reported vaccination status is correct. Non-vaccinated students were recruited at other departments of the student policlinic. The fact that students were using their own smartphones to participate and nobody can be aware of their answers also reduces any social desirability bias, as opposed to putting their answers on paper and then handing the paper to the doctor or nurse. Nevertheless, we clarified this in Limitations.

Changes in text: Discussion, Limitations section, lines 170-175.

While the study identifies religiosity as a factor influencing vaccine uptake, it does not delve into the specific religious teachings or beliefs that may influence this decision. A more nuanced understanding of the intersection between religious beliefs and health behavior is needed.

### We are completely aware of this deficiency of our study, and we agree. This is why we have discussed “religiousness” and underlined that our findings on this topic merely open the door. We further explicated it in the limitations section.

--- Changes in text: in Discussion, Paragraph 3: Nevertheless, in a recent study, Coleman et al. have indicated that single item indices, also used in our study, although most common in the literature, may not be best suited to pro-vide actionable insight regarding the religious beliefs/teachings which influence the vac-cine decision [44]. Therefore, a more detailed, qualitative exploration of this association is warranted.; in Limitations: Qualitative methods could help understand why certain information sources are trusted, providing insights to refine communication strategies, and could also shed light on reli-giousness and religious beliefs and their association with HPV vaccine uptake.

The language writings make the manuscript hard to follow. It should be refined before any further evaluations.

### We have revised the text and are willing to use the English copyeditor.

Round 2

Reviewer 1 Report

Comments and Suggestions for Authors

Thank you for your revision and the clarifications provided. Further clarification is needed though. 

  1. Methods:
    • Could you provide more details on how the hierarchical multivariate logistic regression analysis was conducted? Because your presentation (Table 6) for the regression seems to be confusing. 
    • The variables in Table 6 (presumably the main results) are inconsistent with those presented in Tables 1–5. For instance:
      • In Table 1, "self-reported religiousness" is a continuous variable, whereas in Table 6, "Religiosity" is categorized.
      • Models 2–4 in Table 6 only include certain domains, such as frequency of use. Meanwhile, "Trust" seems to be missing from the regression model.
    • Please provide further elaboration and ensure consistency across the variables in all tables.
  2. Results: The results remain overly detailed and challenging to navigate. Please focus on highlighting the primary results (I guess Table 6 is what readers most interested in) and streamline the presentation of other tables accordingly.
  3. Introduction, Line 25: Regarding the term "HPV vaccine acceptance," do you mean vaccine uptake? As noted, "acceptance" relates to perception, while "uptake" reflects behavior or the actual decision to vaccinate (as you have defined). Please clarify and ensure consistency throughout the manuscript.

Author Response

Thank you for your revision and the clarifications provided. Further clarification is needed though.

Thank you again for reviewing our revised Manuscript and for additional comments and suggestions. We provide answers to your comments and suggestions (denoted by ###) and locations and changes where we have improved our Manuscript (denoted by ---). Of course, we remain open to any additional suggestions as they improve our work.

  • Methods:
    • Could you provide more details on how the hierarchical multivariate logistic regression analysis was conducted? Because your presentation (Table 6) for the regression seems to be confusing. 

### Answer: Table 6, as well as other tables in the manuscript have been refined, and we hope the regression results are now clearer. The hierarchical multivariate logistic regression procedure is described in detail in the Methods section (lines 26-42).

--- Changes in text: Tables 1-6. Methods section, statistical analysis.

  • The variables in Table 6 (presumably the main results) are inconsistent with those presented in Tables 1–5. For instance:
    • In Table 1, "self-reported religiousness" is a continuous variable, whereas in Table 6, "Religiosity" is categorized.

### Answer: Thank you for pointing this out. We have made the term “religiousness” more consistent throughout the text. Also, we explain in the Results section (before the Table 6) that the continuous variable „religiousness“ was categorized into 5 categories, and we provided more comprehensive description.

--- In text: The total score range for the continuous numerical variable “religiousness“ was divided in quintiles for easier interpretation of the results: 0-20 (completely nonreligious), 20.1-40 (moderately nonreligious), 40.1-60 (neither nonreligious nor religious), 60.1-80 (moderately religious), and 80.1-100 (completely religious).“ (lines 6-9).

    • Models 2–4 in Table 6 only include certain domains, such as frequency of use. Meanwhile, "Trust" seems to be missing from the regression model.
    • Please provide further elaboration and ensure consistency across the variables in all tables.

### Answer: Thank you for underlining this issue. When we were building the hierarchical multiple logistic regression models with our statistician, we found, as expected, collinearity between the sources of information and trust in information sources. In addition, adding trust to the model did not yield any additional value to the model, so it was excluded. This information is now added to the Manuscript.

--- Changes in text: In Results, page 9, just below Table 3.

  • Results: The results remain overly detailed and challenging to navigate. Please focus on highlighting the primary results (I guess Table 6 is what readers most interested in) and streamline the presentation of other tables accordingly.

### Answer: All the tables in the manuscript have been refined, and we hope the results they present are now clearer and easier to understand. We have also reduced the overly detailed description of Tables 1-5, focusing on the most important points which are later Discussed, and letting the reader get the other information from the Tables.

--- Changes in text: in Results, description of Tables 1-5.

  • Introduction, Line 25: Regarding the term "HPV vaccine acceptance," do you mean vaccine uptake? As noted, "acceptance" relates to perception, while "uptake" reflects behavior or the actual decision to vaccinate (as you have defined). Please clarify and ensure consistency throughout the manuscript.

### Answer: Thank you for this suggestion, we applied the term “vaccine uptake” consistently throughout the manuscript.

--- Changes in text: Abstract – lines 30, 34 and 39, Introduction - lines 23 and 27, Discussion – line 18

Reviewer 3 Report

Comments and Suggestions for Authors

The language use is much better. And many progress were achieved. But there are still issues to be addressed as follow.

1# The article appears to be akin to an epidemiological survey, which may not fully capture psychological and cognitive aspects due to the lack of relevant research content. Therefore, it is suggested that the authors further enhance the credibility of the information and the degree of information exposure. This could potentially improve the vaccination rate of HPV vaccines in the corresponding regions and among the targeted population. 2# The subjects may hold diverse views on this issue. For instance, some individuals may exhibit field independence or field dependence. Field-independent individuals tend to have strong personal opinions and self-awareness, while field-dependent individuals often rely more on external references. It is unclear whether the article can measure the extent to which an individual is influenced by their peers, or whether their decision to get vaccinated is based on their own subjective will or influenced by others. This variable is not considered in the article, making it difficult to assess the objectivity of the information. 3# Another quality, known as intolerance of uncertainty, also plays a significant role. This refers to an individual's discomfort with the uncertainties in their surroundings, which can diminish their sense of control over their environment, including their health. Such psychological qualities can affect the objectivity and authenticity of the survey analysis. The article fails to take this into account, which is a notable shortcoming. Overall, while the article provides some insights, it needs to address these limitations to strengthen its validity and reliability.

Author Response

The language use is much better. And many progress were achieved. But there are still issues to be addressed as follow.

Thank you again for reviewing our revised Manuscript and for additional comments and suggestions. We provide answers to your comments and suggestions (denoted by ###) and locations and changes where we have improved our Manuscript (denoted by ---). Of course, we remain open to any additional suggestions as they improve our work.

1# The article appears to be akin to an epidemiological survey, which may not fully capture psychological and cognitive aspects due to the lack of relevant research content. Therefore, it is suggested that the authors further enhance the credibility of the information and the degree of information exposure. This could potentially improve the vaccination rate of HPV vaccines in the corresponding regions and among the targeted population.

### Answer: Thank you for this suggestion, we are aware that psychological drivers are important for vaccination decision, however this model is focused on the influence of information environment particularly. Regarding the deeper exploration of the credibility of the information, we suggest in the limitations section that “Qualitative methods could help understand why certain information sources are trusted, providing insights to refine communication strategies“ (lines 184-185)

--- In text: Discussion, Lines 184-185.

2# The subjects may hold diverse views on this issue. For instance, some individuals may exhibit field independence or field dependence. Field-independent individuals tend to have strong personal opinions and self-awareness, while field-dependent individuals often rely more on external references. It is unclear whether the article can measure the extent to which an individual is influenced by their peers, or whether their decision to get vaccinated is based on their own subjective will or influenced by others. This variable is not considered in the article, making it difficult to assess the objectivity of the information.

### Answer: The variables exploring social influence (descriptive and injunctive norms) on vaccination decision were not included in this manuscript, since the model being presented is focused on the influence of media environment on vaccination behavior. We were aware of this limitation, and included it in the manuscript: “Additionally, examining cultural and social factors influencing trust and conducting comparative analyses across regions or countries could uncover unique trends or universal patterns to enhance global health strategies“ (lines 182-184)

--- In text: in Discussion, Lines 182-184.

3# Another quality, known as intolerance of uncertainty, also plays a significant role. This refers to an individual's discomfort with the uncertainties in their surroundings, which can diminish their sense of control over their environment, including their health. Such psychological qualities can affect the objectivity and authenticity of the survey analysis. The article fails to take this into account, which is a notable shortcoming. Overall, while the article provides some insights, it needs to address these limitations to strengthen its validity and reliability.

### Answer: Thank you for your valuable insights, you are absolutely right in your observation that many factors are relevant in determining vaccination behavior. However, we were practically limited regarding manuscript length and word limit, and decided to focus on selected drivers. We plan to analyze wider range of factors in the future, and we explicated these limitations in the manuscript (lines 175-184).

--- In text: In Discussion, Limitations and Future research, Lines 175-184.

Round 3

Reviewer 1 Report

Comments and Suggestions for Authors

No further comments.

Reviewer 3 Report

Comments and Suggestions for Authors

It is acceptable.